# Real-time resolution studies of the regulation of lactate production by hexokinases binding to mitochondria in single cells

**Scott John**[1], **Guillaume Calmettes**[1], **Shili Xu**[2], **Bernard Ribalet**[3]*

**1** Department of Medicine (Division of Cardiology), David Geffen School of Medicine at UCLA, Los Angeles, California, United States of America, **2** California NanoSystems Institute (CNSI) 2151, David Geffen School of Medicine at UCLA, Los Angeles, California, United States of America, **3** Department of Physiology, David Geffen School of Medicine at UCLA, Los Angeles, California, United States of America

* bribalet@mednet.ucla.edu

**Data Availability Statement:** All the data used for this study are added as Excel Files under Supplemental Information.

## Abstract

During hypoxia accumulation of lactate may be a key factor in acidosis-induced tissue damage. Binding of hexokinase (HK) to the outer membrane of mitochondria may have a protective effect under these conditions. We have investigated the regulation of lactate metabolism by hexokinases (HKs), using HEK293 cells in which the endogenous hexokinases have been knocked down to enable overexpression of wild type and mutant HKs. To assess the real-time changes in intracellular lactate levels the cells were also transfected with a lactate specific FRET probe. In the HKI/HKII double knockdown HEK cells, addition of extracellular pyruvate caused a large and sustained decrease in lactate. Upon inhibition of the mitochondrial electron transfer chain by NaCN this effect was reversed as a rapid increase in lactate developed which was followed by a slow and sustained increase in the continued presence of the inhibitor. Incubation of the HKI/HKII double knockdown HEK cells with the inhibitor of the malic enzyme, ME1*, blocked the delayed accumulation of lactate evoked by NaCN. With replacement by overexpression of HKI or HKII the accumulation of intracellular lactate evoked by NaCN was prevented. Blockage of the pentose phosphate pathway with the inhibitor 6-aminonicotinamide (6-AN) abolished the protective effect of HK expression, with NaCN causing again a sustained increase in lactate. The effect of HK was dependent on HK's catalytic activity and interaction with the mitochondrial outer membrane (MOM). Based on these data we propose that transformation of glucose into G6P by HK activates the pentose phosphate pathway which increases the production of NADPH, which then blocks the activity of the malic enzyme to transform malate into pyruvate and lactate.

## Introduction

It is now well accepted that lactate may have beneficial as well as detrimental effects on cell function. For most of the 20th century, lactate was largely considered a dead-end waste product of glycolysis during hypoxia (anaerobic metabolism), causing muscle fatigue [1], and be a key factor in acidosis-induced tissue damage [2,3]. However, since the early 1980s a different

**Funding:** The author(s) received no specific funding for this work.

**Competing interests:** The authors have declared that no competing interests exist.

aspect of lactate metabolism has emerged whereby its transformation into pyruvate may fuel the TCA cycle and provide cellular energy [4–6]. We have studied the metabolism of lactate, and pyruvate, in HEK cells using a cytoplasmic FRET probe specific for lactate which allows temporal and spatial measurements of this substrate in real time in single cells. Using this approach, we have recently shown [7] that pyruvate facilitates the oxidation of lactate by mitochondria via activation of the malate aspartate shuttle (MAS) (Fig 1 left panel), a process that is inhibited by hexokinases (HKs). Under these conditions utilization of lactate as a source of energy would be considered as beneficial for cell metabolism. In contrast, we now report an aspect of lactate metabolism which may be considered detrimental as a result of its accumulation when mitochondria are inhibited (Fig 1 right panel). We will show that this process is also regulated by interaction of catalytically active HKs with the mitochondrial outer membrane (MOM).

### Protection against ischemia-induced acidosis

In many tissues lactate accumulation is detrimental causing acidosis under ischemic (hypoxic) conditions [10]. In this case lactate may originate from different sources as mitochondrial metabolism is inhibited. One source may involve the direct transformation of pyruvate into lactate in the cytosol. This process is enhanced when pyruvate is no longer utilized by mitochondria to form acetyl CoA, following mitochondria inhibition [11]. Another source may involve the reduction of oxaloacetate (OAA) to malate in the mitochondrial matrix, where this reaction uses NADH [12,13]. During ischemia the NADH/NAD ratio increases, due to inhibition of the Electron Transport Chain (ETC). This leads to elevated mitochondrial malate levels as reduction of OAA increases, and as a result, increased flux of malate out of the mitochondria (Fig 1 right panel).

In the cytosol, as part of the pyruvate cycle, the malic enzyme 1 (ME1) facilitates the transformation of malate into pyruvate [9] and subsequently lactate [13]. ME1, or malate dehydrogenase, uses NADP as a cofactor and is a major producer of cytosolic NADPH, which donates high-energy electrons for antioxidant defense and reductive biosynthesis. Cytosolic NADP is recycled into NADPH by ME1, but also primarily by the pentose-phosphate pathway (PPP), and to a lesser extent by isocitrate dehydrogenase 1 (IDH1). In this context it has been recently suggested that the elevated production of NADPH by the HK-dependent glucose 6 phosphate dehydrogenase (G6PD) associated with the PPP inhibits ME1 (Fig 3). This conclusion was based on G6PD knockout experiments showing enhancement, probably as a result of increased NADP, of the flux through ME1 (and IDH1), which led to depletion of cellular malate and citrate [14].

### Interaction of catalytically active HKs with the mitochondrial outer membrane regulates substrate utilization by mitochondria

Most cell types express the two hexokinases I and II (HKI and HKII), but HKI is more abundant in fast proliferating cells and in neurons [15,16], whereas HKII is highly expressed in cells like cardiac and skeletal muscle cells [17]. Many cell lines express a combination of both, with a higher level of HKII in HEK cells. HKI and HKII have different properties. HKII has a higher catalytic activity and binds to the mitochondrial outer membrane (MOM) in a highly dynamic manner, with kinases such as Akt strengthening its interaction with MOM, while G6P (its product) causes its dissociation. In contrast, HKI strongly binds to MOM and cannot be displaced by most metabolic perturbations [18–23].

There is evidence that binding of HKs to the MOM plays a role in regulating substrate utilization by mitochondria by increasing coupling between glycolysis and glucose oxidation,

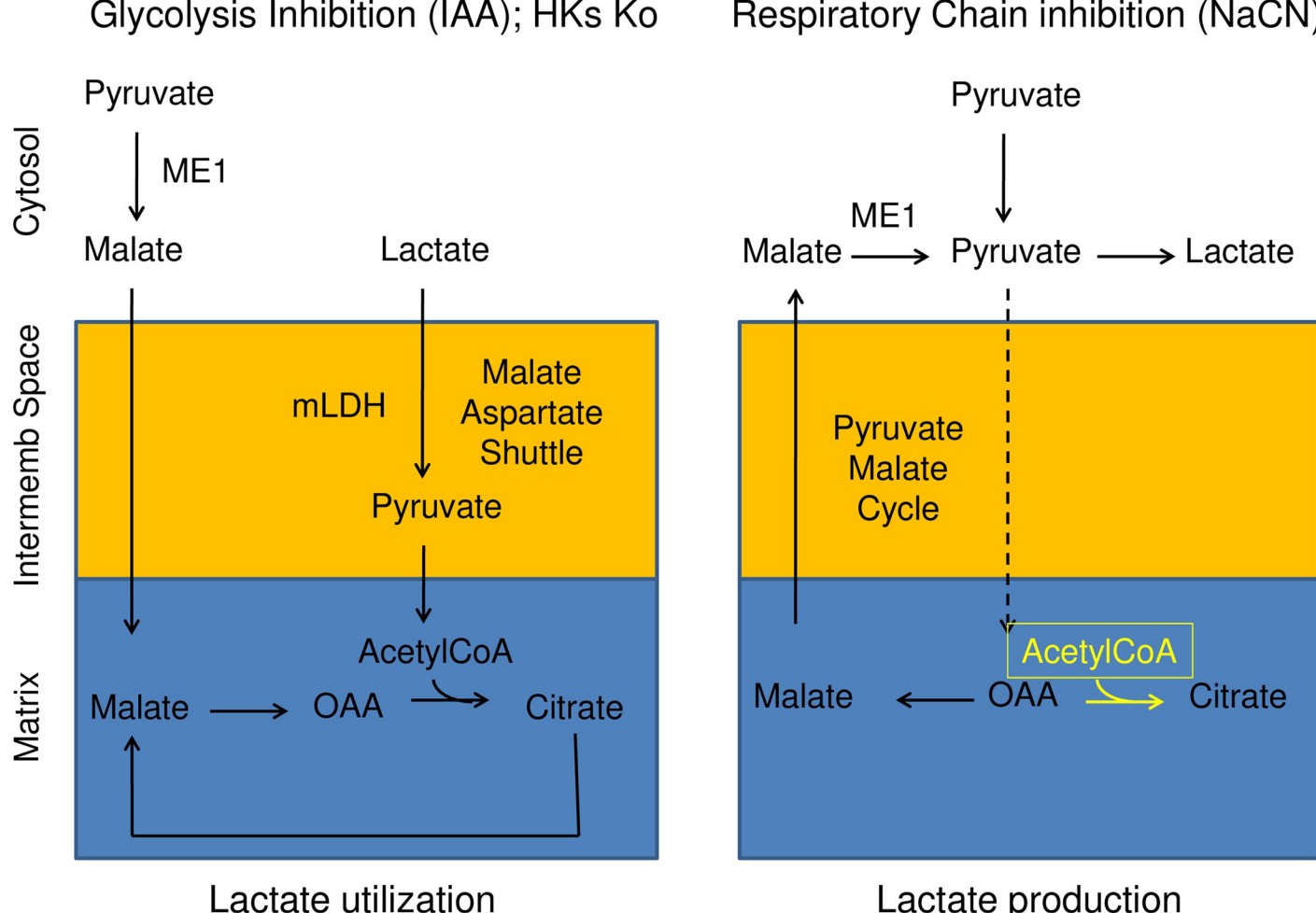

**Fig 1. The cytosolic malic enzyme (ME1) has a dual role depending of the cell's metabolic state.** Left panel. As we recently reported during low energy production by the glycolysis (hexokinase knock down) ME1 may transform pyruvate into malate and then OAA to activate the TCA cycle and may, as a result, transform lactate into pyruvate in the intermembrane space [8]. Right panel. However, under most conditions the ME1 transforms malate, which exits the mitochondria, into pyruvate. In this case ME1 is part of what is known as the pyruvate/malate cycle [9]. When the TCA cycle is inhibited the pyruvate that is generated in this process is no longer taken up by the mitochondria and will be transformed into lactate in the cytosol.

and blocking fatty acid oxidation. For this effect to take place, HKs must be catalytically active [21,24,25]. In addition, binding of catalytically active HKs to the MOM may play an important role in ischemic preconditioning and protection against ischemic injuries [21,23,24,26–30].

We have carried out experiments with the specific lactate FRET sensor "Laconic" to monitor intracellular lactate in real time in HEK in which HKI and HKII could be knocked-down and reintroduced to investigate their roles in regulating the metabolism of pyruvate and lactate. Using this approach, we show (i) lactate accumulation resulting from inhibition of mitochondrial function is prevented by binding of catalytically active HKs to the MOM; (ii) using inhibitors of glucose 6 phosphate dehydrogenase (G6PD) and ME1 that HKs may prevent lactate accumulation following mitochondrial inhibition as a result of NADPH-induced inhibition of ME1. We discuss the putative role this may play in protection against ischemic injuries.

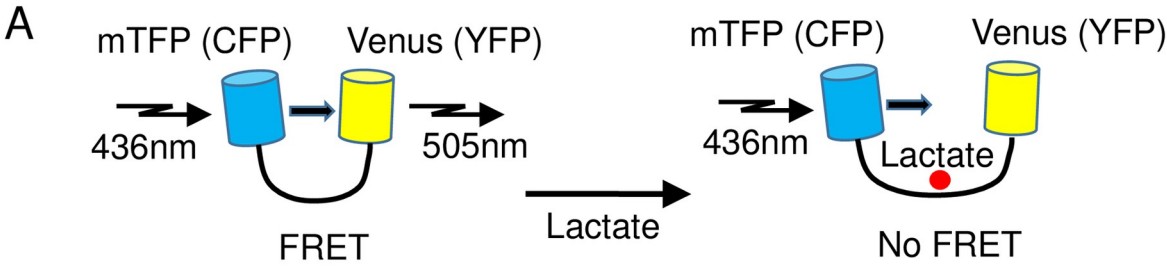

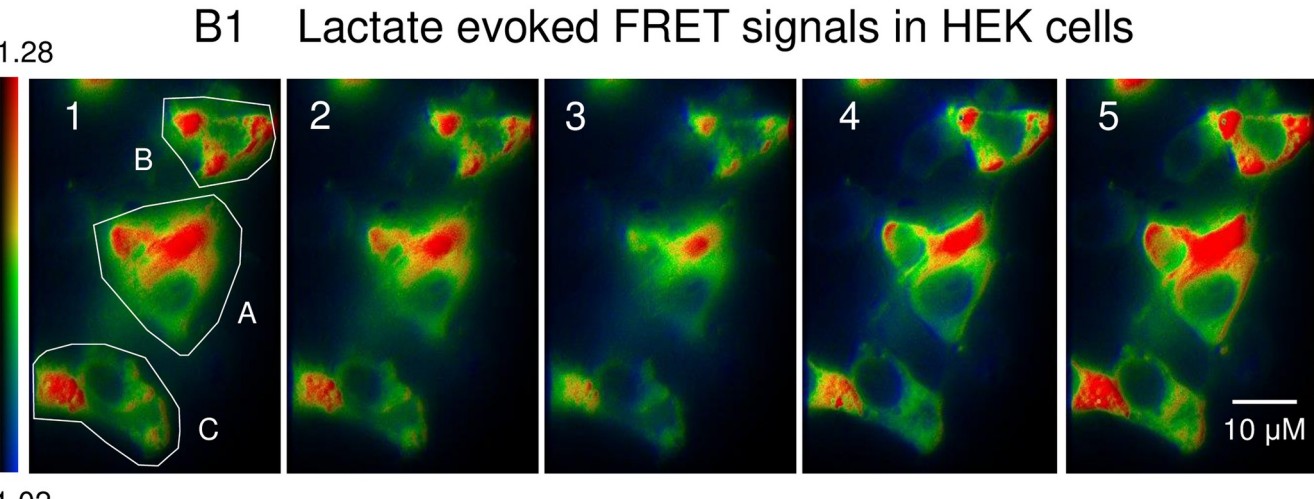

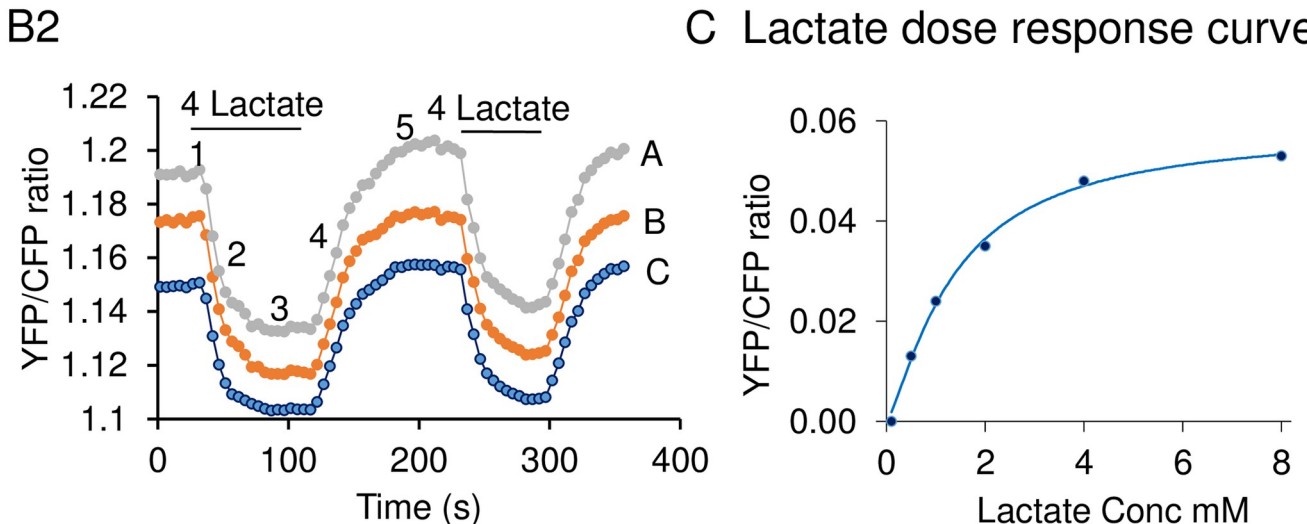

**Fig 2. Methodology.** Panel A. Model of the changes in FRET evoked by lactate binding. In the absence of lactate the donor (CFP) and the acceptor (YFP) are within 1–50 nm of each other and there is significant energy transfer (YFP/CFP ratio is high). Upon binding of lactate YFP and CFP move farther apart, the intensity of YFP fluorescence decreases while that of CFP increases. This leads to a decrease in the YFP/CFP FRET ratio. Panels B1 show images of the changes in FRET ratio evoked by the addition of 4mM lactate. The pseudo color images are shown for illustration purpose, but in most experiments, we only recorded the ratio values shown in panel B2 (not the raw images in panel B1). Every FRET experiment encompassed measurements from four to six

individual cells at the same time in the same microscope field of view. A region of interest (ROI) was drawn around each cell and FRET ratios were recorded in real time over these ROIs. Panel B1 illustrates how addition of lactate, starting at image #1 and ending at image #3, causes a decrease in FRET ratio from 1.28 to 1.02 (see scale bar on the left of panel B1). Upon removal of lactate, starting at image #3 and ending at image #5, the FRET ratio recovered (increase in light intensity). The number of each image corresponds to the numbers on the trace A of panel B2. Panel B2 illustrates how the FRET ratio integrated over the ROI in each cell (shown in panel B1), with varying expression levels of laconic (A, B, C), changes upon addition and removal of lactate applied extracellularly. Here the ratio YFP/CFP is plotted as a function of time. Again, an upward trend of the FRET ratio corresponds to a decrease in intracellular lactate level and vice versa a downward trend indicates an increase in lactate level. These YFP/CFP ratio values were recorded and kept for further analysis. In the experiments presented in this report, data from thirty to hundred cells, taken over separate experiments were used for statistical analysis. Even though the level of expression of the sensor laconic affected the absolute value of the FRET ratio (compare traces A, B, C) the time course of the changes in FRET ratio was similar. This observation suggests that the FRET sensor does not have a strong effect in "buffering" intracellular lactate. Panel C. Graph of the changes in FRET ratio plotted as a function of concentration and fitted with a Hill equation. The fits yield a Kd of 1.26 mM and a Hill coefficient of 1.3. The Kd value obtained from the intracellular milieu is close to that reported by San Martin et al. in 2013 for in vitro experiments (830±160 µM) [31]. Because the FRET ratio values reported in our results are expressed as a percentage of the maximal change obtained at saturating concentration of lactate, these values may be used to get an estimated of the changes in concentration observed under the various experimental conditions.

## Materials and methods

### Solutions and experimental techniques

Most of the techniques used in this study have been described in detail elsewhere [7]. The bath solution for cell imaging consisted of (in mM) 140 NaCl, 5 KCl, 1.1 $MgCl_2$, 2.5 $CaCl_2$, 10 HEPES, with the pH adjusted to 7.2 with NaOH. *N*-methyl-*d*-glucamine was added to maintain the solutions' osmolarity. Five mM glucose was always added to the bath solutions (with or without HK expression), so that HK would be active when present. Solutions were perfused directly over the cells using a gravity-fed eight-way perfusion device (Warner Instruments, Hamden, CT, USA) with electrically controlled solenoids (The Lee Company, Westbrook, CT, USA). Input and output of solution to the recording chamber (glass bottomed Petri dish) were equilibrated to maintain constant flow rates and volumes. Sodium Cyanide (NaCN) and *6*-Aminonicotinamide (*6*-AN) were purchased from Sigma-Aldrich. The inhibitor of ME1 (ME1*) was purchased from (ProbeChem Biochemicals, Shanghai, China).

### HKI and HKII knock down and overexpression in HEK cells. Molecular biology and cell culture

Hexokinase wild-type and mutant constructs were subcloned into the pcDNA3.1 amp mammalian expression vector (Invitrogen), which utilizes the cytomegalovirus promoter. HEK293 cells were transfected with lipofectamine 2000 (Invitrogen). Cells were cultured in Dulbecco's modification of Eagle's medium, high (25 mM) or low (5 mM) glucose medium supplemented with 10% (*v/v*) fetal bovine serum (FBS), penicillin (100 U/ml), streptomycin (100 U/ml), and 2 mM glutamine, and divided twice a week by treatment with trypsin.

Briefly, the approach to DOX-inducible shRNA HK knockdown was as follows: 1) Five shRNA sequences proposed to target specifically either HK1 or HK2 mRNA were located. shHK-1, shHK-2, shHK-3, and shHK-4 targeted the coding sequence of the HK mRNA. shHK-5 targeted the 3' untranslated region of the HK2 mRNA. 2) Sequencing of the shRNAs was performed by Laragen Inc (Culver City, CA). 3) For integration into the genomes, DOX-inducible shRNAs constructs were generated using lentivirus. 4) HK knockdown efficacy of the five shHKs sequences was tested in stable isogenic cells. The successful shRNA were renamed to those demonstrating successful knockdown as reported by John et al 2023 S6 Fig i.e. shHK-1 for HK1 kd and shHK-2 for HK2 kd.

### FRET sensors

Most of the methods used to measure substrate such as lactate are based on enzymatic reactions and have limitations, since they are not able to detect changes in intracellular substrate

**Fig 3. Pyruvate-dependent regulation of lactate metabolism in HEK cells with HKs knocked down.** Elevation of cytosolic pyruvate activates the TCA cycle, which then activates MAS to facilitate the transformation of lactate into pyruvate in the intermembrane space "phase 2". This transformation of lactate into pyruvate is mediated by LDH and the production of NAD by MAS. Pyruvate is then transported into the matrix to further supply the TCA cycle. Inhibition of MAS due to inhibition of the electron transport chain by NaCN stops the transformation of lactate into pyruvate in the intermembrane space, resulting in a fast phase of lactate elevation (phase 4). As shown in Figs 4A1 and 5A this fast phase (phase 4), which is the reversal of phase 2, is followed, in the continued presence of NaCN, by a slower phase of lactate accumulation (phase 5). Two hypotheses may explain this slower phase 5. In one pyruvate may be directly transformed into lactate as the metabolism of pyruvate by mitochondria is blocked. Alternatively, it may involve first the reduction of OAA into malate as the TCA cycle is inhibited and then the transformation of malate into pyruvate, via ME1 in the cytoplasm (phase 5). Lastly pyruvate may be transformed into lactate in the cytoplasm. This schematic also illustrates the effect of the inhibitor of ME1 (ME1*). In this case the experiment was carried out in the absence of HKI, which allows for NaCN-evoked lactate production as a result of ME1 activity. Incubation with the inhibitor ME1* blocks this effect of HKI, thus preventing the development of phase 5. The activity of ME1 in the cytosol is one of the sources of NADPH production. Another source is G6PD the enzyme that mediates the first step of the PPP. The production of NADPH by G6PD may feed back onto ME1 to block the transformation of malate into pyruvate [14]. As a corollary inhibition of G6PD by 6-AN, which prevents the formation of NADPH by PPP would relieve the inhibition of ME1.

levels non-invasively in real-time or with single cell resolution. To circumvent these limitations genetically-encoded reporters have been engineered to monitor with improved spatiotemporal resolution the transport and utilization of substrates, including lactate. Laconic is a genetically-encoded Forster Resonance Energy Transfer (FRET)-based lactate sensor that has been designed with the bacterial transcription factor LldR at its core. This region which forms

the ligand-binding domain is flanked by a pair of fluorescent proteins, CFP and YFP. These two fluorescent proteins have overlapping emission and excitation spectra. Binding of the substrate to the sensor causes a conformational change that affects the relative distance and/or orientation between the two fluorescent proteins, increasing or decreasing FRET efficiency as a result (Fig 2). In the case of laconic the sensor exhibits a nearly linear response between $10^{-5}$ and $10^{-2}$ M and it is not affected by changes in the level of substrates, such as glucose, pyruvate, glutamate, malate or oxaloacetate at cytosolic concentrations [31]. Furthermore, it is insensitive to changes in pH between 7.0 and 7.8 [31,32], ensuring that physiological changes in intracellular pH between 7.0 and 7.4 do not affect the FRET ratio. Plasmid for Laconic was purchased from Addgene (plasmid # 44238) and transfected into HEK 293 cells with lipofectamine 2000 (Invitrogen). Expression of Laconic was sufficiently high after 24 hrs to perform FRET experiments. In our system the probe was expressed in the cytoplasmic compartment, since it has no organelle-targeting sequences associated with it, therefore the Laconic probe reports "global" cytoplasmic lactate levels.

## FRET imaging

The approach to image the cells with the FRET setup has been described in detail elsewhere [7], Briefly, Images (16-bit) were acquired using a Nikon Eclipse TE300 microscope fitted with a 60x (N.A. 1.4) Nikon oil immersion lens and equipped with a filter cube comprising a CFP bandpass excitation filter, 436/20b, together with a longpass dichroic mirror (Chroma Technology Corp, Rockingham, VT, USA). Light-emitting diodes (LEDs, Lumileds, San Jose, CA, USA) were used as light sources: one emitting at 455±20 nm (royal blue) and the other emitting at 505±15 nm (cyan). LEDs and camera exposure were controlled by MetaFluor Imaging 6.1 software (Molecular Devices, Sunnyvale, CA, USA). Ratiometric FRET measurements were performed by simultaneously monitoring CFP and YFP emissions of the sample when excited at the wavelengths for CFP (royal blue LED). The ratio between YFP and CFP emission was measured online in real time using MetaFlour Imaging software. Exposure times were optimized in each case but varied between 100–500 ms. Images were recorded at a constant rate for each cell between 0.2–0.33 Hz.

## Statistical analysis

Most changes in FRET ratio due to various experimental perturbations were fitted with single exponential, a combination of 2 exponentials or a combination of exponential and sigmoid functions (S1C–S1F Fig). The fitted amplitude of the FRET signals was then normalized and expressed as a percentage of control measured with addition of 4 mM lactate (Fig 4A1). To test inhibitors (incubation overnight) or overexpression of HKs we used the same batch of cells (same passage #) for the test and control FRET experiments. Histograms with bin size of 10% were used to display the amplitude of the different phases under the various experimental conditions. Normality and homogeneity of variance were evaluated by Shapiro-Wilk test and F-test two-samples respectively. In the case of non-homogeneity (indicated in figure legends) a t-test two-samples assuming unequal variances was used to evaluate statistical significance. Otherwise a t-test two-samples assuming equal variances was used. Mean ± SEM, standard deviation and P values are reported. A critical value for significance of $P < 0.05$ was used throughout the study. When statistical thresholds of 0.05, 0.005 or 0.0005 are used it is indicated in the figure legends.

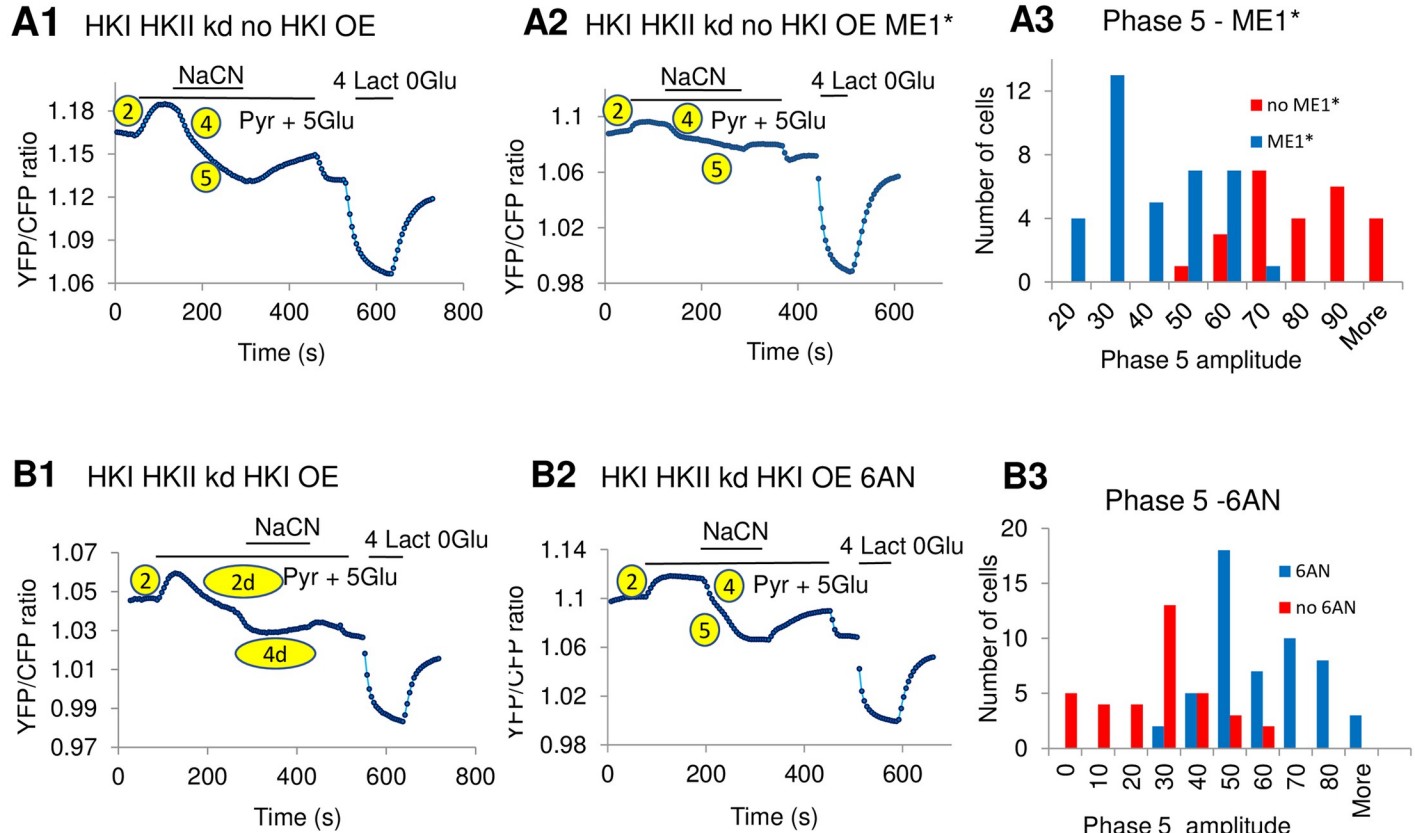

**Fig 4. Inhibitors of the malic enzyme ME1 and of G6PD affect pyruvate-dependent lactate metabolism.** The upper panels A1 to A3 illustrate the effects of HEK cells incubation with the inhibitor of ME1 (ME1*). Panel A1 shows the changes in lactate levels in response to addition of NaCN in the presence of pyruvate, characterized by pronounced phases 4 and 5. In this experiment as in most other experiments, recordings were started in the presence of 5mM glucose. After reaching a steady state YFP/CFP ratio, which usually occurred after a couple of minutes, 2mM pyruvate was added to the perfusion solution. Then NaCN was added once the YFP/CFP ratio had reached a new steady state. This usually occurred within approximately 1min30s. For instance, in panel A1 NaCN was added after 1min34s in the presence of pyruvate + Glucose. In panel A2 NaCN was added after 1min15s in the presence of pyruvate + Glucose. These were typical time intervals needed to reach steady state before the addition of NaCN and lactate. Panel A2 shows the results of a similar addition of NaCN in the presence of pyruvate after incubation of the cells with ME1*. As previously shown the amplitude of phases 2 and 4 decreases by 40% on average [7]. Similarly, the amplitude of phase 5 decreases almost by half, suggesting that inhibiting ME1 with ME1*, which blocks the transformation of malate into pyruvate, prevents the production of lactate evoked by NaCN-induced inhibition of the TCA cycle. Panel A3 shows a quantification of the inhibitory effects of ME1*on the amplitude of phase 5 with a mean phase 5 amplitude of 75.22 ±2.85 (n = 26) without ME1* and a mean amplitude of 35.99±2.32 (n = 38) after incubation with ME1*. The P value in this case was <0.0005. Panels B1 to B3 illustrate how inhibition of G6PD by 6-AN reverses the blocking effect of HKI overexpression (OE). Panel B1 illustrates how overexpression of HKI blocks both phase 2 and 4, which become only transient. There is no longer a phase 5 in this case (compare panels A1 and B1). Panel B2 shows how incubation with 6-AN reverses the inhibitory effect of HKI overexpression with strong recovery of phases 2, 4 and 5. Panel B3 shows a quantification of the effects of 6-AN on the amplitude of phase 5 with a mean amplitude of 22.25±2.80 (n = 37) in the absence of 6AN and 44.43±2.07 (n = 54) after incubation with 6AN. The P value was <0.0005 in this case.

## Results

### Monitoring intracellular lactate levels in single HEK cells with a FRET-based lactate sensor

It is well documented that lactate accumulation during hypoxia is detrimental as it causes acidosis. In fact high intracellular lactate levels with the associated acidosis are good predictors of cardiac tissue damage following ischemia [2]. On this basis we first investigated whether inhibition of mitochondrial oxidative function causes lactate accumulation in our system, and then assessed whether hexokinases could prevent this process. To investigate whether and how HKs may regulate lactate accumulation following mitochondria inhibition, we have carried

out experiments with the FRET-based sensor "Laconic" to specifically monitor the changes in intracellular lactate in HEK cells (Fig 2) in which both HKI and HKII expression had been depleted using shRNAs as indicated in Methods and prior studies [7].

## In the absence of HKs, mitochondrial inhibition causes intracellular lactate accumulation in two phases

We have previously shown that in cells with HKI and HKII knocked down lactate utilization by mitochondria is facilitated by pyruvate [7]. In other words we found that addition of pyruvate leads to a decrease in the cytosolic level of lactate (Fig 1 left panel). We refer to this pyruvate-induced decrease in lactate as phase 2 as illustrated in Figs 4A1 and 5A. In these same cells addition of NaCN reversed the effect of pyruvate by rapidly blocking the utilization of lactate ($\tau = 20.54 \pm 1.45s$). We referred to this inhibitory phase as phase 4. Then, in the continued presence of NaCN, the fast phase of lactate accumulation (phase 4) is followed by a slower phase of lactate accumulation ($\tau = 213.68 \pm 10.40s$). We refer to this slower phase as phase 5 (Figs 4A1 and 5A). We previously hypothesized that the rapid block of lactate utilization by NaCN (phase 4) was due to inhibition of the malate aspartate shuttle (MAS) [7], and we will now explore how mitochondrial inhibition by NaCN may cause the slow phase of lactate accumulation.

## The accumulation of lactate resulting from mitochondrial inhibition depends on the activity of the malic enzyme

During ischemia as the NADH/NAD ratio in the mitochondrial matrix increases oxaloacatate (OAA) is reduced to malate, which in turn may be released into the cytosol. In the cytosol the malic enzyme 1 (ME1) facilitates the transformation of malate into pyruvate and subsequently lactate [13] (Fig 1 right panel and Fig 3). To investigate whether such a mechanism could account for the lactate accumulation observed in our experiments upon mitochondrial inhibition we tested the effects of the ME1 inhibitor ME1* in HEK cells with HKs knocked down. As shown in Fig 4A2 a 24 hrs incubation with ME1* of HEK cells in which HKs had been knocked down markedly depressed the slow increase in lactate accumulation evoked by NaCN (phase 5). We noted that the effect of the malic enzyme inhibitor ME1* is not always consistent (see histogram in panel 4A3). This may be due in part to the instability of the compound. Nonetheless, the quantification shown in panel 4A3 supports the hypothesis that addition of NaCN in ME1* treated HEK cells, with HKI and HKII knocked down, suppresses the sustained increase in intracellular lactate (phase 5). These results suggest that in the absence of HK, the accumulation of lactate induced by mitochondrial inhibition involves the activity of ME1, the production of malate by mitochondria, its transformation into pyruvate and subsequently lactate.

## The transformation of malate into pyruvate by ME1 is inhibited by the PPP in an HK-dependent manner

We next investigated how HKs may block the large and sustained increase in lactate (phase 5) that occurs upon inhibition of the electron transport chain (ETC). HK produces G6P, and the subsequent utilization of this product as a substrate by G6PD to generate NADPH, blocks ME1-mediated conversion of malate into pyruvate [14]. Accordingly, in the presence of HKs, the accumulation of lactate due to the transformation of pyruvate into lactate as a result of the inhibition of the ETC by NaCN would not take place. To test this hypothesis we investigated how 6-AN, an inhibitor of G6PD may affect the production of lactate in HEK cells

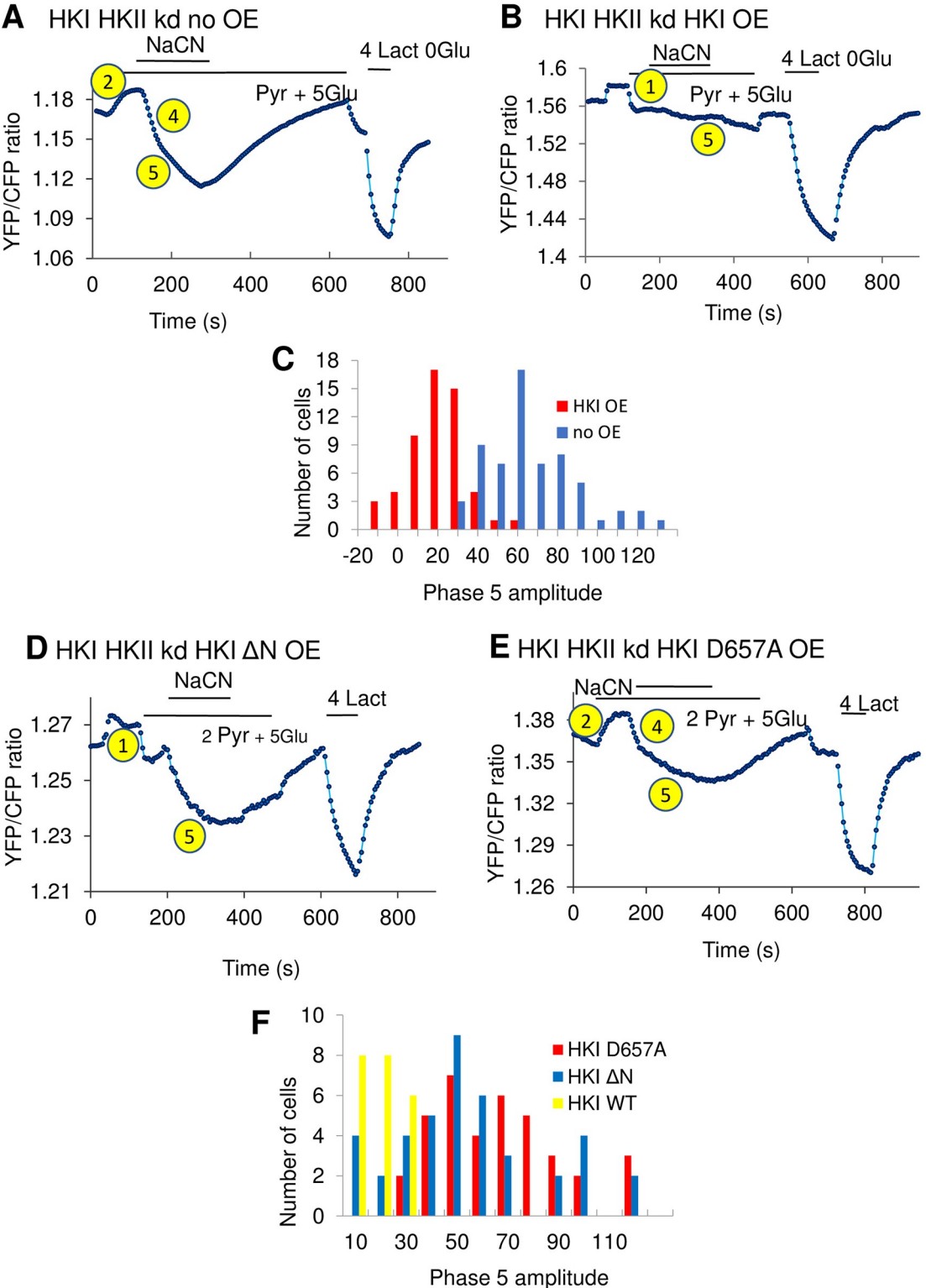

**Fig 5. Modulation of pyruvate-dependent lactate metabolism by overexpression of HKI, the catalytically inactive D657A mutant and the truncated ΔN mutant.** The FRET signal in panel A is similar to that in panel A1 Fig 4 and illustrates the time course of phases 2, 4 and 5. In this case addition of NaCN, which was applied 1min35s after exposure to pyruvate, causes an increase in lactate in both phase 4 and 5. Upon removal of lactate there is a recovery which is strongly dependent upon the presence of pyruvate. Panel B illustrates one of the effects of wild type HKI overexpression (OE) as previously reported. The first

striking difference with panel A is the sharp and fast increase in lactate, which occurred within 20s following addition of pyruvate. This increase which can be assimilated to a sustained phase 1 is likely due to the transformation of pyruvate into lactate in the cytoplasm. Following this rapid increase in lactate, there was no distinctive phase 4 or 5 in response to NaCN. The histogram in panel C is a quantification of this effect. The mean phase 5 amplitude with no HKI overexpression was 64.39±8.11 (n = 63) and 16.05±1.88 (n = 54). The P value was <0.0005. Data in panel D illustrate the effects of HKI ΔN overexpression. In many cells a clear phase 5 was observed upon addition of NaCN, indicating that in this case overexpression of the HKI ΔN did not block the increase in lactate evoked by NaCN. Data in panel E illustrate the effects of overexpression of the catalytically inactive HKI D657A. In this case the results were very clear, the FRET signal was not affected. The amplitude and time course of the response resemble that obtained in the absence of HKI expression (panel A) with clear phases 2, 4 and 5. The histograms in panel F show a quantification of the effects of wild type and mutants HKI on the amplitude of phase 5. The mean phase 5 amplitude was 49.97 ±4.62 (n = 41) for HKI ΔN compared to 15.71±2.35 (n = 22) for HKI wt overexpression. The P value was <0.0005. The mean phase 5 amplitude was 63.55±4.02 (n = 37) for HKI D657A overexpression compared to 15.71±2.35 (n = 22) for HKI wt overexpression. In this case P<0.0005.

expressing HKI. For these experiments, cells expressing HKI were incubated with 6-AN for 24 hrs for maximal effects. Under these conditions the response to addition of pyruvate and then NaCN was robust. Indeed, as shown in Fig 4B2, incubation with 6-AN prevented the blocking effect of HKI, and a strong slow phase of lactate production (phase 5) was elicited upon addition of NaCN. This slow phase followed the reactivation of the initial decrease in lactate utilization (phase 2) as previously reported [7]. Altogether these data support the hypothesis whereby the production of NADPH by HKs-dependent activation of the PPP prevents the increased production of lactate generated by the activity of ME1 after mitochondrial inhibition.

## To block lactate accumulation (phase 5) HKs must be catalytically active and must interact with MOM

To test whether binding of catalytically active HKs to MOM is necessary for the regulation of the metabolism of lactate by mitochondria we overexpressed two mutants of HK, one with the first 11 residues deleted (ΔN) and the catalytically inactive HKI D657A [20,33]. Importantly, data that we had previously obtained with CHO cells showed that the D657A mutation of HKI, which blocks the enzyme's catalytic activity, does not affect its interaction with MOM, while truncating the first 11 aa of the N terminus (HKI ΔN) prevented the interaction of HKs with MOM without altering its catalytic activity (Figs 5 and 6).

For reference, data in the upper panels of Fig 5 shows the inhibitory effects of wild-type HKI characterized by a lack of phases 2 and 5. In this case there is no phase 2 and only a phase 1 is observed. Panel D illustrates the effects of HKI ΔN. When compared to the trace obtained with wild-type HKI (Panel B) we clearly see that addition of NaCN to HEK cells expressing HKI ΔN evokes an augmentation of intracellular lactate (phase 5). This result suggests that truncation of HKI N terminus prevents HKs-dependent modulation of lactate metabolism by mitochondria. Thus, binding of HKs to MOM may play a role in regulating the metabolism of lactate. It should be noted that HKI ΔN did occasionally block the effects of pyruvate and NaCN due perhaps to overexpression (see Figs S4 and 6).

Data in Fig 5E show that HKs must be catalytically active to regulate lactate metabolism by mitochondria. In this case overexpression of the catalytically inactive HKI D657A mutant did not block the initial decrease in intracellular lactate level evoked by pyruvate (phase 2) and did not prevent lactate accumulation evoked by NaCN (phase 5). This lack of effect of HKI D657A was observed despite the fact that this mutant HK did interact with MOM (Fig 6).

To further test the hypothesis that HKs must be catalytically active to prevent lactate accumulation, we carried out experiments with wild-type HEK cells in the absence of glucose. Data in Fig 7A show that removal of glucose has the same effect as the catalytically inactive HK

# Distribution of HKI-YFP wild-type and mutants in CHO cells

### HKI-YFP expressed in CHO cells

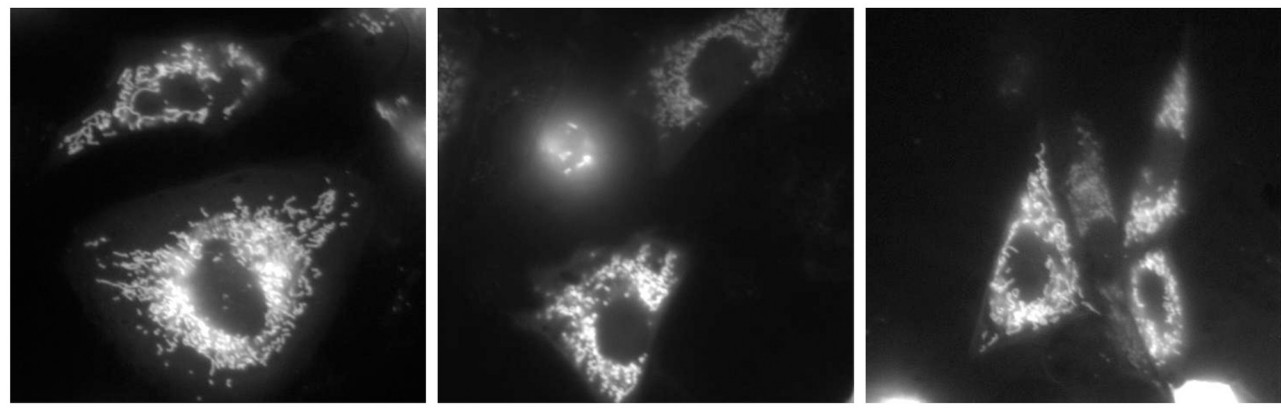

### HKI D657A-YFP expressed in CHO cells

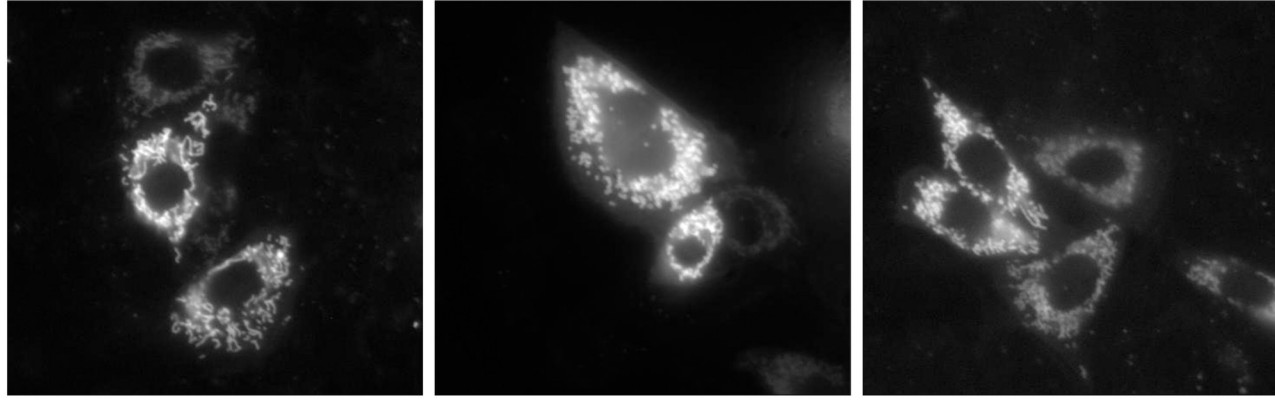

### HKI ΔN-YFP expressed in CHO cells

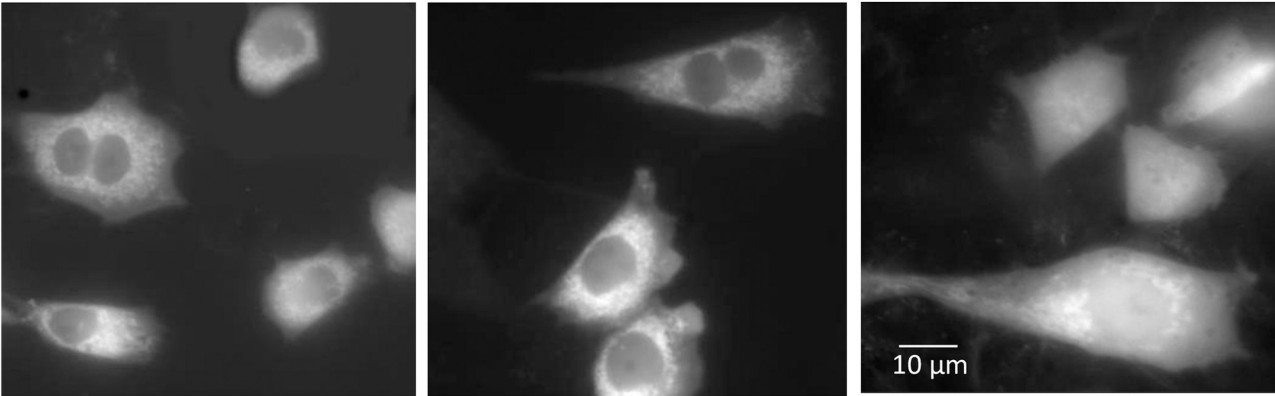

10 μm

**Fig 6. Intracellular localization of wt HKI-YFP, HKI D657A-YFP and HKI ΔN-YFP expressed in CHO cells.** Even though the intracellular distribution of HKI tagged with YFP at the N terminus (HKI-YFP) was similar in HEK and CHO cells we show images using CHO cells because the cells are slightly larger, which renders the images clearer. Images in the upper panel show that HKI-YFP interacts strongly with the outer mitochondrial membrane. Similarly the catalytically inactive HKI D657A-YFP binds to the outer mitochondrial membrane. In contrast deletion of the N terminus of HKI-YFP (HKI ΔN-YFP) prevents the interaction of the truncated HKI. However, as shown in some of the images there is still some HKI binding to the

mitochondria. Although HKI ΔN blocked NaCN-induced changes in pyruvate and lactate metabolism in few cases, there was absolutely no effect of HK ΔN in other cases. We suggest that in the former cases binding HKI ΔN to MOM is due to its overexpression. This does not weaken the argument in favor of a requirement for HK binding to MOM under physiological levels of expression.

D657A or the knocking down of HKI/HKII. Indeed addition of NaCN in the absence of glucose causes production of lactate with prominent phases 4 and 5. Together these observations suggest that both the binding to OMM and the catalytic activity of HKs are necessary for HK-dependent regulation of lactate metabolism by mitochondria.

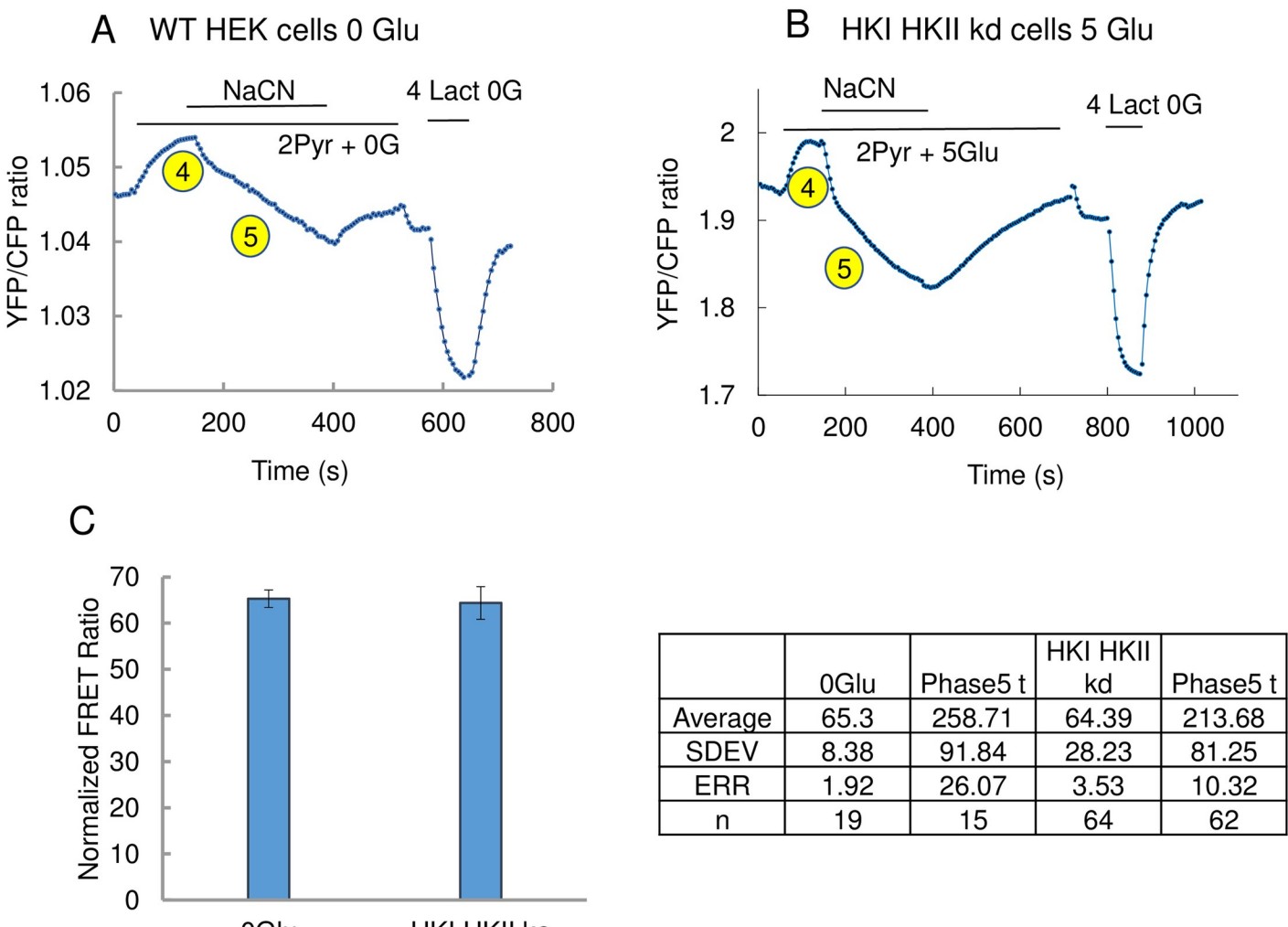

**Fig 7.** Addition of NaCN increases intracellular lactate level in wild-type HEK cells (A) and HKI-HKII kd HEK cells (B). Panel A. Effects of NaCN addition in WT HEK cells in the absence of glucose. Starting with the beginning of the trace, addition of 2mM pyruvate caused a rapid and sustained decrease in intracellular lactate. This decrease is rapidly reversed upon addition of NaCN. This early effect of NaCN is referred to as phase 4. This phase is then followed by a slow and sustained increase in lactate in the continued presence of NaCN. This slow phase is referred to as phase 5. Panel B illustrates a similar experiment performed with HEK cells in which HKI and HKII had been knocked down. The experiment carried out, in this case, in the presence of 5mM glucose shows a similar effect, with NaCN addition causing an increase in lactate in two phases (a fast phase "4" and a slow phase "5"). Panel C is a graph illustrating the difference in phase 5 amplitude in wild type (0 glucose) and HKI-HKII kd (5mM glucose). There is no statistical difference between the effect of NaCN in the 2 cases. The table on the right of the figure shows the values used to generate the graph.

## Discussion

We have used engineered HEK cells to study the regulation of lactate metabolism by hexokinases (HKs). To this end both HKI and HKII could be knocked down in a DOX-dependent manner. HKI or HKII could then be reintroduced one at a time. The cells were also transfected with a FRET-based sensor to specifically monitor rapid changes in intracellular lactate level in real-time in single cells.

The main observations that we have made are: 1) In the absence of HK expression pyruvate-induced decrease in cytoplasmic lactate is rapidly blocked by the electron transporter chain (ETC) inhibitor NaCN. In the continued presence of NaCN a slow and sustained accumulation of lactate develops. 2) Incubation with the malic enzyme inhibitor ME1* blocks the effects of NaCN. 3) Overexpression of HKs inhibits both the fast and sustained increase in lactate evoked by the addition of NaCN. 4) The inhibition of NaCN-induced lactate accumulation by HKs is reversed by incubation with the PPP inhibitor 6-AN.

### The protective role of hexokinases against lactate accumulation

Lactate utilization by mitochondria may be beneficial to fuel the cell's energy requirement [34]. Our previous work supports this observation [7]. However, it may also have adverse effects and causes acidosis under hypoxic conditions as its level rises. Hence it is a good predictor of cardiac tissue damage following ischemia [2,3]. Noteworthy, it has been reported that interaction of catalytically active HKs with MOM may be protective and prevent the deleterious effects of lactate accumulation. This proposal is supported in part by data obtained with the TAT peptide, which prevents interaction of HK's N terminal with MOM [22,29]. Our data agree with this premise as we have shown that intracellular lactate accumulation evoked by mitochondrial inhibition occurs in the absence of HKs and is blocked by overexpression of either HKI or HKII. Furthermore, we have shown that this protective effect requires HKs to be catalytically active and to be interacting with MOM. These observations establish the relevance of our system to study the regulation of lactate metabolism by HKs.

### Accumulation of lactate evoked by mitochondrial inhibition involves the generation of malate

Inhibition of mitochondrial metabolism as a result of ischemia or of poisoning may lead to lactate accumulation due to the transformation of pyruvate into lactate in the cytoplasm when pyruvate is no longer used by mitochondria. Although we show that lactate accumulation evoked by mitochondrial inhibition occurs in the presence of pyruvate our data indicate that a different mechanism contributes to pyruvate-dependent elevation of intracellular lactate. Under normal respiratory conditions malate is transformed into oxaloacetate (OAA) as part of the TCA cycle. However, increase in the NADH/NAD ratio due to inhibition of the ETC may lead to the reduction of oxaloacetate and production of malate in the mitochondrial matrix, where this reaction is facilitated by NADH [12,13]. As the concentration of malate rises inside the matrix, efflux of malate out of the mitochondria will take place. Once in the cytoplasm malate will in turn be transformed into pyruvate by the malic enzyme ME1. This reaction is part of the pyruvate cycle as reported in pancreatic beta cells [9] (see also Fig 1). However, as reutilization of pyruvate by mitochondria decreases in the presence of inhibitors of the respiration, pyruvate transformation will give rise to elevated lactate level, as the pyruvate/lactate equilibrium favors the formation of lactate in the cytoplasm [11,35]. We have used an inhibitor of ME1, ME1*, to test the hypothesis that the elevation of lactate induced by NaCN in HKI/

HKII knocked down cells involves ME1 activity. Our results show that incubation with ME1* for over 24 hrs blocks NaCN-induced lactate accumulation under our experimental conditions. These data support the hypothesis that the malic enzyme ME1 is involved in the generation of cytosolic lactate following mitochondrial inhibition.

## HKs suppression of NaCN-induced lactate accumulation involves NADH production by the PPP

It has been recently suggested that the production of NADPH by G6PD feeds back negatively onto ME1, the activity of which is also coupled to the transformation of NADP into NADPH, to prevent the transformation of malate into pyruvate [14] and subsequently lactate [13]. We have tested whether such a process could account for the effect of HK on NaCN-induced lactate accumulation in our system. Our data show that 24–48hr treatment of HEK cells expressing HKI with a specific inhibitor of G6PD, 6-AN, prevents the blocking of NaCN-induced lactate accumulation by HKs. In other words, after treatment with 6-AN, NaCN application evokes lactate accumulation even in the presence of HKs. Based on these observations we propose that reversal of the effect of HK by 6-AN is due to the decreased production of NADPH by PPP and reactivation of ME1. These data together with the effect of the ME1 inhibitor (ME1*) in the absence of HK support our hypothesis whereby the effect of HK on NaCN-induced lactate accumulation involves in part PPP-induced NADPH production which feeds back negatively onto ME1, effectively lowering malate to pyruvate conversion and thus lactate production.

## HKs interaction with MOM

It has been known for many years that HKs bind to the mitochondria outer membrane (MOM) with HKI binding more strongly than HKII [18–20]. Furthermore, binding of HKs to MOM is enhanced during ischemia and this may play an important role in protection against ischemia-induced injuries [21,22,36], where HKs must be catalytically active for protection [33]. Our data showing that the regulation of lactate metabolism requires HK's catalytic activity and binding to MOM is consistent with this hypothesis and suggest that such regulation of lactate metabolism by HKs may be involved in protection against ischemia.

## Conclusions and future directions

Our studies have previously shown that lactate is used by mitochondria and that this process requires activation of the TCA cycle by pyruvate. We now show that inhibition of mitochondrial function causes a large increase in lactate level as a result of malate reduction in the cytoplasm. This process is counteracted by HKs which activates the production of NADPH by PPP and subsequently inhibits ME1. Such an effect of HKs requires binding of the catalytically active enzyme to MOM. We propose that this regulation of cytosolic lactate by HKs may play a protecting role against ischemic injury.

The pathways that we have tested involve enzymes such as G6PD and ME1 that have already been knocked out in cultured cells. We want to use this approach to further characterize the role of these enzymes in the metabolism of pyruvate and lactate. In particular, we would like to ascertain the role of G6PD in mediating the effects of HKs, and the dual role of ME1 in regulating the reversible reaction, depicted in Fig 1, between malate and pyruvate as a function of the cell metabolic state.

## Supporting information

**S1 Fig. Time course of changes in FRET ratios, and curve fitting of the experimental traces.** Panels A and B show traces taken from panels A1 and A2 in Fig 4, on an extended time scale, to emphasize the time course of the effect of Pyruvate and NaCN in Hki/HKII Kd cells in the presence and absence of ME1*. Panels C, D, E and F show curve fitting of the changes in FRET ratio. In this figure and all the other figures a downward trend in FRET ratio indicates an increase in intracellular lactate. In panel C the increase in intracellular lactate levels evoked by NaCN (phases 4 and 5) was fitted with a sum of two exponential functions. The amplitudes and time constants of the two phases were derived from the fit. In D the changes evoked by the inhibitor 6AN was best fitted using a combination of exponential and sigmoidal functions. In this case the amplitude of phase 5 was derived from the difference between the calculated maximum and minimum. In E and F the effects of NaCN on the amplitude of phases 2 and 4 was estimated as the difference between the beginning and end of the fitted traces. When the value at end of the trace exceeded that at the beginning a negative value for the amplitude was derived from the calculation.
(TIF)

**S2 Fig. Effects of HKI and HKII knock down and overexpression (OE) in HEK cells.** The upper panel depicts the effects of knocking down (kd) HKII and HKI HKII in HEK cells. In HKI HKII kd (no OE) Mean phase 5 normalized amplitude was 64.4 ± 3.55 (n = 63). In HKII kd (no OE) Mean phase 5 normalized amplitude was 71.6 ± 5.08 (n = 21). F-test yielded a P value of 0.12. A t-test (equal variance) P value was 0.97. The shapiro-wilk test P value for HKI HKII kd no OE was 0.1, and 0.07 for HKII. The lower panel depicts the effects of the overexpression of HKI and of HKII in HEK cells in which HKI and HKII had been previously knocked down. In HKI HKII kd cells, HKI OE Mean phase 5 normalized amplitude was 16 ±1.8 (n = 55). In HKI HKII kd cells, HKII OE Mean phase 5 normalized amplitude was 23.4 ±3.8 (n = 27). An F-test yielded a P value of 0.012 and a t-test (unequal variance) yielded P<0.0005. The shapiro-wilk test P value for HKI OE was 0.07, and 0.9 for HKII OE.
(TIF)

**S3 Fig. Time constant for the changes in FRET ratio evoked by addition of pyruvate, AOA and NaCN.** Phase 2t1 is for the addition of pyruvate, phase 3t1 for the addition of AOA, Phases 4t1 and 5t1 for the addition of NaCN and lactate t1 for the addition of lactate. Mean 33.98207 15.79144 20.53946 213.6842 21.87948. SDEV 13.85233 6.90242 11.35711 81.24961 7.461619. ERR Mean 1.74523 0.71193 1.45413 10.40295 0.940076.
(TIF)

**S4 Fig. The dual effects of ΔN HKI overexpression.** Data in panels A shows the effects of NaCN addition in HKI HKII kd cells for reference. Clear Phases 2, 4 and 5 are observed in this case. Panels B and C show two very different results observed with ΔN HKI overexpression (OE). In one case (panel B) overexpression of the mutant HKI had effects very similar to those of the wild type with only a phase 1 and no phases 4 and 5. However, in other cells a clear phase 5 was observed (Panel C). Thus, in this case overexpression of the ΔN HKI did not block the increase in lactate evoked by NaCN. This dichotomy is consistent with the imaging data obtained with ΔN HKI-YFP (Fig 6).
(TIF)

**S5 Fig. Data obtained in experiments carried out in HEK cells in which HKI and HKII were knocked down and in which HKI was overexpressed.** In this Excel sheet, as with the following Excel sheets, the first column corresponds to the cell number. In columns "Phase 2a,

3a, 4a and 5a" the values correspond to the amplitude of phase 2, 3, 4 and 5 derived from the fit of the FRET traces. In columns "Phase 2t1, 3t1, 4t1 and 5t1" the values correspond to the time constant of Phase 2, 3, 4 and 5 derived from the fit of the FRET traces. Values in the columns labeled "lactate a" and "lactate t1" correspond to the amplitude and time constant of the FRET signal generated by the addition of 4mM lactate. Values in columns labeled "Norm 2a, 3a, 4a and 5a" were generated by normalizing the values for 2a, 3a, 4a and 5a to their corresponding value obtained upon addition of lactate in the same cell. Each row in this spread sheet correspond to one cell. Whole numbers for the time constants result from a fit of the data that exceeded the limits provided during the fit.
(XLSX)

**S6 Fig. Data obtained in experiments carried out in HEK cells in which HKI and HKII were knocked down and in which HKII was overexpressed.**
(XLSX)

**S7 Fig. Data obtained in experiments carried out in HEK cells in which HKI had been overexpressed and which have been incubated for 24hrs in the presence of 6-Aminonicotinamide (6-AN).**
(XLSX)

**S8 Fig. Data in this worksheet were used as a control for the experimental data presented in supplemental Fig 7.** In this case the cells were not incubated in the presence of 6-Aminonicotinamide (6-AN). Experimental data obtained with and without 6-AN were recorded on the same day.
(XLSX)

## Author Contributions

**Conceptualization:** Scott John, Guillaume Calmettes, Shili Xu, Bernard Ribalet.

**Data curation:** Scott John, Shili Xu, Bernard Ribalet.

**Formal analysis:** Guillaume Calmettes, Bernard Ribalet.

**Methodology:** Bernard Ribalet.

**Project administration:** Bernard Ribalet.

**Resources:** Bernard Ribalet.

**Writing – original draft:** Bernard Ribalet.

**Writing – review & editing:** Scott John, Guillaume Calmettes, Shili Xu, Bernard Ribalet.

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
