## [Decision Letter · Decision Letter 0]

4 Jan 2024

PONE-D-23-37473Real-time resolution studies of the regulation of lactate production by hexokinases binding to mitochondria in single cells.PLOS ONE

Dear Dr. Ribalet,

Thank you for submitting your manuscript to PLOS ONE. After careful consideration, we feel that it has merit but does not fully meet PLOS ONE’s publication criteria as it currently stands. Therefore, we invite you to submit a revised version of the manuscript that addresses the points raised during the review process.

Please submit your revised manuscript by Feb 18 2024 11:59PM. If you will need more time than this to complete your revisions, please reply to this message or contact the journal office at plosone@plos.org. Please include the following items when submitting your revised manuscript:A rebuttal letter that responds to each point raised by the academic editor and reviewer(s). You should upload this letter as a separate file labeled 'Response to Reviewers'.A marked-up copy of your manuscript that highlights changes made to the original version. You should upload this as a separate file labeled 'Revised Manuscript with Track Changes'.An unmarked version of your revised paper without tracked changes. You should upload this as a separate file labeled 'Manuscript'.

We look forward to receiving your revised manuscript.

Kind regards,

Bashir Sajo Mienda, PhD

Academic Editor

PLOS ONE

Journal Requirements:

Reviewers' comments:

Reviewer's Responses to Questions

**Comments to the Author**

1. Is the manuscript technically sound, and do the data support the conclusions?

Reviewer #1: Yes

Reviewer #2: Yes

2. Has the statistical analysis been performed appropriately and rigorously? 

Reviewer #1: Yes

Reviewer #2: N/A

3. Have the authors made all data underlying the findings in their manuscript fully available?

Reviewer #1: Yes

Reviewer #2: Yes

4. Is the manuscript presented in an intelligible fashion and written in standard English?

Reviewer #1: Yes

Reviewer #2: Yes

5. Review Comments to the Author

Reviewer #1: The manuscript by John et al. entitled “Real-time resolution studies of the regulation of lactate production by hexokinases binding to mitochondria in single cells” reports on the effects triggered by hexokinases I and II on the production of lactate by human HEK293 cells fed with pyruvate and glucose, and thereafter exposed to cyanide. Overall, the manuscript is of general interest and shows the outcome of a series of elegant experiments performed with analytical skill. Moreover, it is my opinion that the work by John et al. features sufficient novelty when compared to recent studies on the regulation of lactate production in human cells, which is a point of great relevance for a deeper understanding of the energetic metabolism of cancer cells. In particular, the present manuscript provides data that nicely deepen the recent observation recently published by the Authors (John et al., PLoS ONE, 18: e0286660). Accordingly, I strongly recommend the publication in PLoS ONE of the manuscript by John et al. Nevertheless, I would recommend to the Authors some amendments to their manuscript before publication:

Major points:

1) I suggest to modify the Figures showing the kinetics of lactate production. In particular, I recommend to omit the experimental observations detected over the 500-800 s time- interval. This would let the use of a scale for the dependent variable (e.g. from 1.12 to 1.19 for panel A1 of Figure 4) more suitable to represent the relevant kinetic phases shown. In my opinion what the Authors state at page 15 (Methods, statistical analysis): “The fitted amplitude of the FRET signals was then normalized and expressed as a percentage of control measured with addition of 4 mM lactate (Fig. 1B).” does suffice to inform the Readers about the way the data were normalized (anybody interested in checking this can have a look at the Excel files linked to the manuscript). Eventually, the Authors could include in the Figures continuous lines representing the fittings obtained to interpret the different kinetic phases.

2) Concerning the previous point: at page 15 it is mentioned that Figure 1 contain Panels showing some fittings (Fig. 1C-E) and the normalization procedure (Fig. 1B). Honestly, I cannot see these Panels in Figure 1. Something was missed? Please amend.

3) Page 18, legend of Figure 4: I agree with the Authors that the inhibition of malic enzyme does negatively affect phase 5 (during which lactate is produced at a slow rate) and I also agree with them that this is an important finding. However, I disagree with the suggestion that “phase 5 is almost completely gone”. The legend of Figure 4 does indeed state that the amplitude of this phase was halved (from 75 to 36), i.e. that it was not almost completely suppressed. Moreover, to what extent were affected phases 2 and 4 in the same experiment? Please amend this part of the manuscript.

4) What the Authors mean with “time constant”? Are they referring to a rate constant? What units were used to express this variable?

Minor points:

a) Abstract: please specify that OMM denotes Outer Mitochondrial Membrane.

b) I suggest to replace “mitochondrial inhibition by NaCN” with “inhibition of mitochondrial electron transfer by NaCN”.

c) The Authors use quite frequently the term “activation” referred to different enzymes (i.e. malic enzyme, lactate dehydrogenase). I suggest to specify what “activation” indicates.

d) Legend of Figure 1: “exists” should be replaced with “exits”?

e) In my opinion, the technique used by the Authors to monitor lactate in vivo is very elegant. Accordingly, it would be of importance to mention over which pH interval the laconic sensor does efficiently monitor lactate concentration (this could be mentioned in page 13, in the “FRET sensors” paragraph where the work of Machler et al. and that of San Martín et al. is mentioned). This would help the Readers interested to use this probe to know under which acidic conditions the laconic sensor could be appropriately used.

f) Legend of Figure 2: I would replace “YFP intensity decreases” with “the intensity of YFP fluorescence decreases”. The same of course for CFP.

g) Figure 2: please specify the parametric form of the Hill equation fitted to the experimental observations shown in Panel C.

h) I suggest to include in the legends of Figures 4,5, and 7 the timing of the different additions (after how many seconds pyruvate and glucose, and NaCN were added)

i) Page 16: “For reference, the FRET probe Laconic comprises a binding domain which upon interaction with lactate enables energy transfer between two fluorophores (CFP and YFP) that flank the substrate binding site.”. It might be the contrary (as it seems in Figure2)?

j) Page 20: I suggest to replace “catalytic active HKs” with “catalytically active HKs”.

k) Introduction and Results sections contain some repetitions. Please consider to shorten the manuscript accordingly.

Reviewer #2: In the manuscript submitted by John et al., the authors described an interesting work, in which the underlying biochemical mechanism of lactate production by hexokinases under different conditions were investigated. Although the topic as well as the results are of interest, there are several issues should be addressed:

1. There is no page or line number, thus hard to have comments specified to certain paragraph or sentences.

2. Materials and Methods: HK1 and HK2 knock down, the shRNAs described were all for HK2. Also, the authors stated that knock-down efficiency was shown in Fig. 1A, but there is no such information in Fig. 1.

3. The choose of cell model is in doubt. The authors talked about ischemia injury to the heart, but the cells used were kidney cells. Do they act the same as heart cells? It is better to use heart cells, or, to talk about ischemia damage to the kidney in the text.

4. Results part I, much of the information had been described in Materials and Methods, and thus could be omitted.

5. Fig. 4, legend, lactate metabolism is NOT regulated by inhibitors, but by the enzymes. Inhibitors are the tools to prove that the enzymes regulate the metabolism.

6. Fig. 6, the authors claimed that HKI interacts strongly with the OMM, however, it is hard to tell based on the images provided. The authors at least, should also label the mitochondria and take confocal images to show the interaction. Otherwise, it can only be said that these proteins are distributed in the cytosol.

7. Some typos. For example, “first step” was written as “fist step” (in Results section, page with Fig. 3, bottom part). Many more are present, please check.

6. PLOS authors have the option to publish the peer review history of their article (what does this mean?). If published, this will include your full peer review and any attached files.

Reviewer #1: No

Reviewer #2: **Yes: **Jun Yang

---

## [Author Response · Author response to Decision Letter 0]

25 Jan 2024

Response to the Reviewers’ comments.

We would like to thank the Reviewers for their thoughtful and constructive comments.

We have removed the parts that were repetitive and shortened the manuscript accordingly. These changes are highlighted in yellow and are “strikethrough” in the marked-up version. The marked up version details also the changes that have been added to address the reviewers’ comments and are highlighted in blue.

Detailed response to Reviewer #1 comments

Major points:

1) I suggest to modify the Figures showing the kinetics of lactate production. In particular, I recommend to omit the experimental observations detected over the 500-800 s time- interval. This would let the use of a scale for the dependent variable (e.g. from 1.12 to 1.19 for panel A1 of Figure 4) more suitable to represent the relevant kinetic phases shown. In my opinion what the Authors state at page 15 (Methods, statistical analysis): “The fitted amplitude of the FRET signals was then normalized and expressed as a percentage of control measured with addition of 4 mM lactate (Fig. 1B).” does suffice to inform the Readers about the way the data were normalized (anybody interested in checking this can have a look at the Excel files linked to the manuscript). Eventually, the Authors could include in the Figures continuous lines representing the fittings obtained to interpret the different kinetic phases.

We understand the reviewer’s comment regarding the expansion of the time scale and the removal of the lactate reference. This is what we had done in our previous paper (John et al. 2023) (figures 3 and 4) to better appreciate the changes in time course in phases 2 and 4.

Following the reviewer’s comment we removed, as shown in Fig 1S (upper panels), the 4mM lactate reference from two traces of Fig 4 and expanded the remaining of the traces. In doing so the time course of the fast phases were better illustrated. However, the visual appreciation of the difference in amplitude of the slow phase (phase 5) was lost and one could conclude just by looking at the traces that ME1* did not have any effect. It is true that the reader should look at the scale to appreciate the effect of ME1*, but this may not often be the case. We believe that having the 4mM lactate reference helps visualizing the effects of the experimental manipulations on the amplitude of phase 5.

We also tried to expand short sections of the traces showing the time course of phases 2 and 4 and insert these short segments in the main figures 4 and 5. We can do it, but it does not seem appropriate since i) this data is already published and ii) the present manuscript focuses on the slow phase 5.

We are open to suggestions.

Fits of the data (Phases 4 and 5, 2d and 4d) have been included in fig 1S. 

2) Concerning the previous point: at page 15 it is mentioned that Figure 1 contain Panels showing some fittings (Fig. 1C-E) and the normalization procedure (Fig. 1B). Honestly, I cannot see these Panels in Figure 1. Something was missed? Please amend.

Yes, the fits were missing. They are now added to fig 1S.

3) Page 18, legend of Figure 4: I agree with the Authors that the inhibition of malic enzyme does negatively affect phase 5 (during which lactate is produced at a slow rate) and I also agree with them that this is an important finding. However, I disagree with the suggestion that “phase 5 is almost completely gone”. The legend of Figure 4 does indeed state that the amplitude of this phase was halved (from 75 to 36), i.e. that it was not almost completely suppressed. Moreover, to what extent were affected phases 2 and 4 in the same experiment? Please amend this part of the manuscript.

This is correct, we were overly zealous! Indeed, the amplitude decreases by half. The text has been corrected accordingly (see blue highlights). We also refer now to the decrease in amplitude of phases 2 and 4, which are also halved.

4) What the Authors mean with “time constant”? Are they referring to a rate constant? What units were used to express this variable?

The equation used to fit some of the data is f=a*exp(-t/τ). In this case the time constant τ is expressed in s. The unit has been added in the text when these values are presented.

Minor points:

a) Abstract: please specify that OMM denotes Outer Mitochondrial Membrane.

This is now specified throughout the text. We use MOM, but it should not make a difference.

b) I suggest to replace “mitochondrial inhibition by NaCN” with “inhibition of mitochondrial electron transfer by NaCN”.

This is done we use “inhibition of the electron transport chain” throughout the text.

c) The Authors use quite frequently the term “activation” referred to different enzymes (i.e. malic enzyme, lactate dehydrogenase). I suggest to specify what “activation” indicates.

We no longer use the word “activation” to refer to the enzyme. For example a sentence would now read: “In the cytosol activation of the malic enzyme 1 (ME1) facilitates the transformation of malate into pyruvate.”

d) Legend of Figure 1: “exists” should be replaced with “exits”?

Corrected

e) In my opinion, the technique used by the Authors to monitor lactate in vivo is very elegant. Accordingly, it would be of importance to mention over which pH interval the laconic sensor does efficiently monitor lactate concentration (this could be mentioned in page 13, in the “FRET sensors” paragraph where the work of Machler et al. and that of San Martín et al. is mentioned). This would help the Readers interested to use this probe to know under which acidic conditions the laconic sensor could be appropriately used.

This is now mentioned when we describe the probe under FRET sensor.

f) Legend of Figure 2: I would replace “YFP intensity decreases” with “the intensity of YFP fluorescence decreases”. The same of course for CFP.

Corrected.

g) Figure 2: please specify the parametric form of the Hill equation fitted to the experimental observations shown in Panel C.

This has been added.

h) I suggest to include in the legends of Figures 4,5, and 7 the timing of the different additions (after how many seconds pyruvate and glucose, and NaCN were added).

The timing for the addition of NaCN and pyruvate is now added in the legend of figures 4 and 5.

i) Page 16: “For reference, the FRET probe Laconic comprises a binding domain which upon interaction with lactate enables energy transfer between two fluorophores (CFP and YFP) that flank the substrate binding site”. It might be the contrary (as it seems in Figure2)?

Yes, it is. This has now been corrected.

j) Page 20: I suggest to replace “catalytic active HKs” with “catalytically active HKs”.

k) Introduction and Results sections contain some repetitions. Please consider to shorten the manuscript accordingly.

We now use “catalytically active” throughout the text.

Detailed response to Reviewer #2 comments

Reviewer #2: In the manuscript submitted by John et al., the authors described an interesting work, in which the underlying biochemical mechanism of lactate production by hexokinases under different conditions were investigated. Although the topic as well as the results are of interest, there are several issues should be addressed:

1. There is no page or line number, thus hard to have comments specified to certain paragraph or sentences.

Page number and line number are now added.

2. Materials and Methods: HK1 and HK2 knock down, the shRNAs described were all for HK2. Also, the authors stated that knock-down efficiency was shown in Fig. 1A, but there is no such information in Fig. 1.

This part of the Methods regarding the knock down of HK1 and HK2 was indeed very confusing and has been rewritten.

Regarding Fig 1A. We had the information regarding the knock-down efficiency in the original manuscript, but since this data was originally published in John et al. 2023, we omitted it from the present manuscript. The reference to Fig 1A is now omitted from the present manuscript and we refer to the previous publication.

3. The choice of cell model is in doubt. The authors talked about ischemia injury to the heart, but the cells used were kidney cells. Do they act the same as heart cells? It is better to use heart cells, or, to talk about ischemia damage to the kidney in the text.

This is a very good point. We now refer to ischemia in general.

We often refer to cardiac cells in the manuscript because it is one system that we have used extensively. However, we have used the lactate probe in other cell types such as Hela cells, neonatal cardiac myocytes and IPS cells.

Our goal in the presence studies was to identify the pathways that are responsible for the production and utilization of lactate. This study took a long time, even with the use of HEK cells that are easy to manipulate.

Our goal now is to knock down ME1, G6PD and GOT1 to further support our hypotheses.

In addition, we plan to investigate these processes in adult cardiac myocytes. However, to carry out this study we need to generate a number of probes using adenovirus vectors. We have the technique to do it, but it will take more than a year and some support to achieve this goal.

4. Results part I, much of the information had been described in Materials and Methods, and thus could be omitted.

Yes, this was repetitious and we have shortened the manuscript significantly for that reason.

5. Fig. 4, legend, lactate metabolism is NOT regulated by inhibitors, but by the enzymes. Inhibitors are the tools to prove that the enzymes regulate the metabolism.

We have made the correction

6. Fig. 6, the authors claimed that HKI interacts strongly with the OMM, however, it is hard to tell based on the images provided. The authors at least, should also label the mitochondria and take confocal images to show the interaction. Otherwise, it can only be said that these proteins are distributed in the cytosol.

This is a reasonably well established finding. We (Calmettes et al. 2015; John et al. 2011) as well as others ( Nederlof et al., 2013; Southworth, Davey, Warley, & Garlick, 2007; Sui & Wilson, 1997; Wilson, 2003) have studied in detail the interaction of HKs with OMM and reported the strong interaction of HKI with OMM. Our experiments in adult and neonatal cardiac myocytes (Calmettes et al., 2015) were carried out using confocal imaging. Colocalization experiments with mitochondrial markers were also performed.

7. Some typos. For example, “first step” was written as “fist step” (in Results section, page with Fig. 3, bottom part). Many more are present, please check.

We have gone over the manuscript to make sure that there are no typos.

---

## [Decision Letter · Decision Letter 1]

23 Feb 2024

Real-time resolution studies of the regulation of lactate production by hexokinases binding to mitochondria in single cells.

PONE-D-23-37473R1

Dear Dr. RIBALET,

We’re pleased to inform you that your manuscript has been judged scientifically suitable for publication and will be formally accepted for publication once it meets all outstanding technical requirements.

Kind regards,

Bashir Sajo Mienda, PhD

Academic Editor

PLOS ONE

Additional Editor Comments (optional):

Reviewers' comments:

Reviewer's Responses to Questions

**Comments to the Author**

1. If the authors have adequately addressed your comments raised in a previous round of review and you feel that this manuscript is now acceptable for publication, you may indicate that here to bypass the “Comments to the Author” section, enter your conflict of interest statement in the “Confidential to Editor” section, and submit your "Accept" recommendation.

Reviewer #1: All comments have been addressed

Reviewer #2: All comments have been addressed

2. Is the manuscript technically sound, and do the data support the conclusions?

Reviewer #1: Yes

Reviewer #2: Yes

3. Has the statistical analysis been performed appropriately and rigorously? 

Reviewer #1: Yes

Reviewer #2: N/A

4. Have the authors made all data underlying the findings in their manuscript fully available?

Reviewer #1: Yes

Reviewer #2: Yes

5. Is the manuscript presented in an intelligible fashion and written in standard English?

Reviewer #1: Yes

Reviewer #2: Yes

6. Review Comments to the Author

Reviewer #1: The Authors addressed all the points I raised with my previous review. Accordingly, I recommend the publication of this manuscript in PLoS ONE.

Reviewer #2: The authors have adequately addressed the issues raised by this reviewer. No further questions regarding this submission.

7. PLOS authors have the option to publish the peer review history of their article (what does this mean?). If published, this will include your full peer review and any attached files.

Reviewer #1: **Yes: **Alejandro Hochkoeppler

Reviewer #2: No

---

## [Editor Report · Acceptance letter]

27 Feb 2024

PONE-D-23-37473R1 

PLOS ONE

Dear Dr. Ribalet, 

I'm pleased to inform you that your manuscript has been deemed suitable for publication in PLOS ONE. Congratulations! Your manuscript is now being handed over to our production team.

Kind regards, 

on behalf of

Dr. Bashir Sajo Mienda 

Academic Editor

PLOS ONE